# Global Internet Data on the Interest in Antibiotics and Probiotics Generated by Google Trends

**DOI:** 10.3390/antibiotics8030147

**Published:** 2019-09-12

**Authors:** Kamiński Mikołaj, Łoniewski Igor, Marlicz Wojciech

**Affiliations:** 1Sanprobi Sp.z.o.o., 70-535 Szczecin, Poland; 2Faculty of Medicine I, Poznan University of Medical Sciences, 60-780 Poznań, Poland; 3Department of Biochemistry and Human Nutrition, Pomeranian Medical University, 70-204 Szczecin, Poland; 4Department of Gastroenterology, Pomeranian Medical University, 70-204 Szczecin, Poland

**Keywords:** antibiotics, probiotics, Google Trends, Internet, antimicrobial resistance

## Abstract

Data from the Google search engine enables the assessment of Google users’ interest in a specific topic. We analyzed the world trends in searches associated with the topics “antibiotics” and “probiotics” from January 2004 to June 2019, using Google Trends. We analyzed the yearly trends and seasonal variation. We performed an R-Spearman rank correlation analysis of the relative search volume (RSV) of the topics in 2015 with antibiotic consumption, health expenditure per capita, and the 2015 Human Development Index (HDI) of the country. The mean interest in the topic of antibiotics was equal to RSV = 57.5 ± 17.9, rising by 3.7 RSV/year (6.5%/year), while that of probiotics was RSV = 14.1 ± 7.9, which rose by 1.7 RSV/year (12.1%). The seasonal amplitude of antibiotics was equal to RSV = 9.8, while probiotics was RSV = 2.7. The seasonal peaks for both topics were observed in the cold months. The RSV of probiotics, but not antibiotics, was associated with antibiotic consumption (Rs = 0.35; *p* < 0.01), health expenditure (Rs = 0.41; *p* < 0.001), and HDI (Rs = 0.44; *p* < 0.001). Google users’ interest in antibiotic- and probiotic-related information increases from year to year, and peaks in cold months. The interest in probiotic-related information might be associated with antibiotic consumption, health expenditure, and the development status of the Google users’ country.

## 1. Introduction

The data on the efficacy of probiotics is growing [1], and is generating interest among scientists, clinicians, and patients [2]. Currently, trial results may be quickly forwarded by social media to a wide audience of conscious consumers [3]. However, access to probiotics may be limited by regulatory agencies, and lack of knowledge on the beneficial properties of microbes in the local population [3]. The interest in probiotics may be seasonal. Some changes over time, which may be associated with the seasonal variation of gastrointestinal ailments, reveal a seasonal variation with a higher prevalence in autumn–winter [4,5]. Moreover, it is reported that the consumption of antibiotics peaks in winter and decreases in summer [6,7]. These seasonal variations may also be associated with an increase in the number of probiotic recommendations to relieve gastrointestinal complaints, or to restore gut microbiota. Indeed, we observed that the sales of probiotics are higher in colder than in warm months (Łoniewski and Marlicz-unpublished data). Up to 90% of Web users look for health-related information [8]. The Internet provides immediate access to an enormous amount of information. Beck et al. reported that ~80% of users perceive the Web as a reliable source of information [9]. It is suggested that Internet traffic and areas of interest may mirror the health issues of the population [10]. The most popular search engine globally is Google. Some past studies have used Google data for the epidemiologic analysis of health-related issues [11,12,13]. We assume that Google data may reflect the changes in global interest in the terms and subjects related to antibiotics and probiotics. Therefore, we aimed to analyze, with a decomposition of time series data, to present the trends over time, seasonal variation, and irregular variation.

We aimed to investigate the interest (from 2004 to 2019) in the search terms “antibiotics”, “antibiotic resistance”, and “probiotics” among Google users worldwide.

## 2. Methods

We collected the data on 7 June 2019 using the Google tool, Google Trends (GT; https://trends.google.com/trends/). GT is an open-source tool that presents the relative search volume (RSV) of a specific phrase in a given country and period in the Google search engine. The data is available from January 2004 to the day of the query. Moreover, GT recognizes many “topics” that enables us to analyze the search volume of the same topic in all languages, for example, car and umbrella. It facilitates the easy comparison of the given terms across the world. RSV is an index of search volume adjusted to the number of Google users in a given geographical area. RSV ranges from 1–100, where the value of 100 indicates the peak of popularity (100% of popularity in the given period and location) and 0 the nadir (<1%) [10]. GT also enables the comparison of more than one phrase or topic simultaneously. In this case, RSV = 100 corresponds to the highest popularity of one of the compared terms. Initially, we chose the following topics: “antibiotics”, “probiotics”, “antimicrobial resistance”, “prebiotics”, and “synbiotics”. To assess the ratio of interest in the topics, we calculated the ratio of the mean RSV of all of the topics to antibiotics. Moreover, we calculated a similar ratio of the number of findings in the PubMed search engine for all of the topics to antibiotics from 2004 to the date of collection. Because the interest in the topics may vary significantly, we included topics in the secondary analysis, with mean RSVs equal to or higher than 5. We compared the RSV of the chosen terms worldwide from January 2004 to the day of the collection. We excluded countries with low search volumes from the analysis. To ensure that the research is reproducible, we followed the recommendations of Nuti et al., and presented the search conditions in a modified checklist (Appendix A) [10]. We analyzed the popularity of the selected topics using the Seasonal Decomposition of Time Series by Loess (Local Polynomial Regression Fitting), included in the *ggseas* package of R 3.6.1. (R Foundation, Vienna, Austria) [14]. The analysis decomposes time series data to present trends over time, including seasonal and irregular variations. The trend component reflects the long-term progression of the series, the seasonal variation presents the periodic changes of the time series, and the irregular variation presents the residuals of the time series after the other components are removed, representing random factors. All of the obtained components of the time series are presented in the same units as the primal time series, which, in our study, denotes RSV (*y*-axis) over time (*x*-axis). Most Google users live in the northern hemisphere, thus most of them experience their cold season from December to February. We chose three countries (Argentina, Australia, and South Africa) from the southern hemisphere to perform an additional time series decomposition, in order to reveal possible inverse RSV seasonal patterns related to the difference in months of the cold season. Additionally, we analyzed the time series decomposition of RSV of several countries with the highest relative interest for “probiotics”. We illustrated the results using the *ggplot* and *ggseas* R packages [15]. We used the Seasonal Mann–Kendall test to detect the presence of a significant trend in the analyzed time trends (α = 0.05) [16]. For significant trends, we performed a linear regression to estimate the slope expressed as changes in RSV per year, and the percentage of mean RSV per year. To determine the occurrence of the significant seasonal component, we fitted an exponential smoothing state space model with a Box-Cox transformation, autoregressive-moving average errors, and trend and seasonal components (TBATS) to the data [17]. In the case of a significant seasonal component, we calculated the seasonal amplitude by adding the absolute values of the highest and lowest monthly seasonal components, and expressed them as RSV and percentage of mean RSV. A *p*-value < 0.05 was considered a significant difference.

GT also presents a comparison of the chosen term in sub-regions. All of the chosen topics in a given sub-region added up to 100%. We set countries as sub-regions. We did not illustrate the RSV from the sub-region comparison, and presented them in a grey color. The RSV was presented as a mean ± standard deviation or percentage in the country analysis. Moreover, we used the R-Spearman rank correlation to search for associations between the RSV of the topics antibiotics and probiotics, with antibiotic consumption [18], antibiotic resistance (measured by the Drug Resistance Index) [19], inflation-adjusted healthcare expenditure per capita [20], and the Human Development Index (HDI) of the countries [21]. DRI represents the combined ability of antibiotics to treat infections, with the extent of their use in clinical circumstances, for a specific region [19]. Inflation-adjusted health expenditure per capita is a universal measure of the sum of the private and public health funding. The outcome is adjusted to the 2011 dollar. The Human Development Index is a combined index of life expectancy, education, and per capita income, which allows for comparing the development of the countries. Because the most recent and detailed information on antibiotics consumption and antibiotics resistance available are for 2015, we extracted the data for 2015 from GT for all of the countries with a significant RSV.

## 3. Results

In the initial analysis with all five topics, the mean RSV of antibiotics was equal to 57.6 ± 18.0, probiotic RSV = 14.1 ± 7.9, antimicrobial resistance RSV = 1.7 ± 0.6, prebiotic RSV = 1.3 ± 0.7, and synbiotics RSV <1 (which could not be accurately compared with the other topics). The ratio of Google users’ interest was as follows: 4.1 (antibiotics/probiotic), 33.9 (antibiotics/antimicrobial resistance), and 44.3 (antibiotics/prebiotic). The ratios of PubMed findings were as follows: 16.9 (antibiotics/probiotic), 2.4 (antibiotics/antimicrobial resistance), and 56.6 (antibiotics/prebiotic). In the secondary analysis with two topics, antibiotics generated the highest interest in February 2019 (RSV = 100), and the lowest in July 2006 (RSV = 33), with a mean RSV of 57.5 ± 17.9. Probiotics presented the highest interest in March 2019 (RSV = 33), and the lowest in fourteen months in 2004–2006 (RSV = 5), with a mean RSV of 14.1 ± 7.9 (Table 1). The visualization of the seasonal decomposition is presented in Figure 1.

The highest relative interest in probiotic was noted in Taiwan (probiotic/antibiotics; 58%/42%), Greece (45%/55%), Slovakia (36/64%), Hong Kong (34%/66%), Canada (33%/66%), Singapore (33%/67%), Serbia (32%/68%), Bulgaria (31%/69%), Hungary (31%/69%), and the USA (31%/69%) (Figure 2), while the highest relative interest in antibiotics was observed in Egypt (5%/95%), Japan (5%/95%), Russia (5%/95%), Saudi Arabia (5%/95%), Vietnam (5%/95%), Kazakhstan (6%/94%), Venezuela (6%/94%), Argentina (8%/92%), Austria (9%/91%), Germany (9%/91%), and the Ukraine (9%/91%) (Appendix A).

We observed peaks in interest in the topics in June (Argentina—antibiotics and probiotics), August (Australia and South Africa—antibiotics), and September (Australia—probiotics), which corresponds to the local cold seasons (Table 1). Probiotics did not have any significant seasonal variation in South Africa. The time series decompositions of the topics for Argentina, Australia, and South Africa are presented in Appendix A. We analyzed the time series decomposition of RSV of Taiwan, Greece, Canada, and the United States (Appendix A).

We found significant associations between the RSVs of probiotic and antibiotic consumption, inflation-adjusted health expenditure per capita, and HDI (Table 2). There were no significant associations for antibiotics.

## 4. Discussion

We found that Google users’ interest in antibiotics is on average four times higher than for probiotics. The interest in antimicrobial resistance, prebiotics, and synbiotics was marginal in comparison with antibiotics and probiotics. Interestingly, the number of findings generated in the PubMed search engine for the years 2004–2019 for the phrase antibiotics is ~2.4 times higher than for the phrase antimicrobial resistance, while in Google Trends, the ratio of RSV between both topics were equal, at ~34. These differences may reflect the lack of public interest in antimicrobial resistance and the need for increasing public awareness of this issue [22,23,24].

The global antibiotic consumption increased by 39% between 2000 and 2015, and it is estimated that it will be up to 200% higher in 2030 than in 2015 [25]. The probiotics market is growing, and is speculated to grow 37% globally from 2016 to 2020 [26]. The RSV of both antibiotics and probiotics is increasing over time. This might be associated with the growing consumption of both types of products. Additionally, the growing number of studies and the regulation of antibiotics and probiotics may generate public discourse. However, since 1990, no new classes of antibiotics were discovered. We found that the RSV of the topic probiotics increased relatively twice as fast as antibiotics. This may be related to the wide range of new beneficial effects of probiotics in the prevention of surgery-related complications [27], mental health [28], the cardiovascular system [29,30], insulin resistance [31], atopy [32], and many others. Furthermore, because of regulations, probiotics can be publicly advertised, while antibiotics cannot. It is also suggested that the community is aware of antibiotics, but have a little knowledge on their properties, risks and antimicrobial resistance [33].

Van Boeckel et al. reported that antibiotic consumption is at its highest during the cold season globally [6]. Moreover, the peaks of the number of antibiotic-related searches may be associated with the increased prevalence of infections in the cold season. Simultaneously, probiotics may be recommended to prevent antibiotic-associated diarrhea [34,35]. The fact that the nadir of both topics occurs in different months is also worth discussing. The decrease in the number of infections in summer may be related to the decrease in interest in the topic of antibiotics. Surprisingly, the nadir of the interest in the topic probiotics occurs in December, whereas in this month, both interest in and the consumption of antibiotics increases. We could not find any plausible explanation for this phenomenon. Another notable finding in the present study was the high variation in the interest in the topics antibiotics and probiotics among the countries. Interestingly, only in Taiwan was the RSV of the topic probiotics higher than antibiotics. The highest interest in probiotics was observed in several Balkans countries, the USA, and Canada, and minor highly developed countries in the Far East. Inversely, the regions with the highest interest in antibiotic-related information did not present a geographic pattern. The association between the RSV of the topic probiotics and antibiotic consumption might be caused by the recommendations of probiotics for the prevention of antibiotic-associated diarrhea, and to restore gut microbiota after antimicrobial treatment [34]. Google users from more developed countries generated more searches related to the topic probiotics. We speculate that it might be associated with better public awareness and the development of the probiotic market. We found that the RSV of the topic probiotics was positively associated with antibiotic consumption, health expenditure per capita, and HDI in 2015. This may be associated with the development of the local probiotics industry in the wealthier countries. Interestingly, the RSV of the topic antibiotics was not significantly associated with antibiotic consumption. The interest of antibiotics may not mirror the antibiotic consumption, as a result of the health policies limiting access to this type of drug. Therefore, the interest and will of a Google user cannot be easily converted into the purchase. Moreover, antibiotic consumption is highly dependent on the in-hospital use of antibiotics [36,37], we hypothesize that hospitalized patients may not generate Internet traffic related to the topic antibiotics. Antibiotics represent the old type of drug, and are used by clinicians all over the world. Therefore, the interest in the topic antibiotics may not correspond with health expenditure per capita and HDI. The DRI was not significantly associated with interest in antibiotics. It may be explained by no relation between the RSV of the topic and antibiotics consumption, which could potentially generate an increase in antibiotic resistance. However, the analysis was performed only for 2015 which limits the conclusions.

This study was the first of its kind to explore the interest of Internet users in antibiotics, probiotics, prebiotics, synbiotics, and antimicrobial resistance. We followed the recommendation of Nuti et al., so as to ensure the reproducibility of the study [10]. We found that the users’ interest corresponds to the epidemiological data on antibiotic consumption. Moreover, we found that the topic antimicrobial resistance may be underrepresented in Internet discourse. Finally, we found that the interest in probiotics-related information varied among the countries, but is positively associated with the development status of the country, health expenditure, and antibiotic consumption.

The authors acknowledge several limitations of this study. Firstly, GT only enables the estimation of RSV; it is not possible to use GT to assess a precise number of queries. Secondly, using GT is the most credible for relatively common phenomena, which are resistant to media clamour [38]. The RSV of the topics could be dependent on media attention, and this could be the reason for the high amplitude of the irregular component in the time series of the topics in Argentina, South Africa, and Australia. Thirdly, the results are limited because of the low search volume in many, mostly African, countries. Fourthly, because of limited data on antibiotic consumption, we could only perform the R Spearman correlation test for 2015. Finally, using GT may provide insight into an under-researched epidemiological pattern, but the observations should be verified with real-world research.

Taken together, our research reveals that Google users’ interest in antibiotic- and probiotic-related information increases from year to year, and peaks in cold months. Internet traffic related to the topic probiotics doubled in comparison with the topic antibiotics. The interest in probiotic-related information is associated with antibiotic consumption, health expenditure, and the development status of the Google users’ country.

## Figures and Tables

**Figure 1 antibiotics-08-00147-f001:**
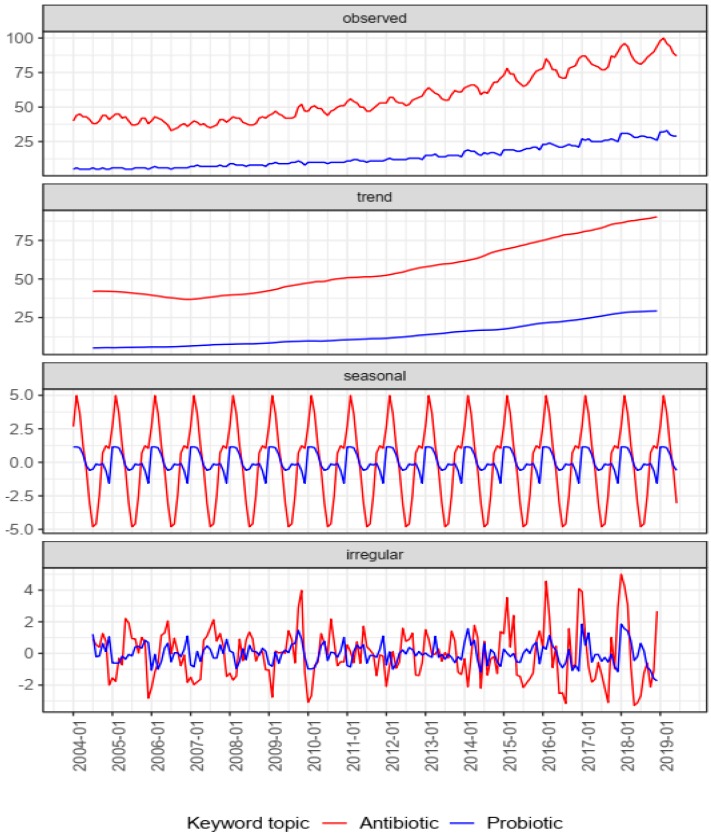
Relative search volume (RSV) of topics “antibiotic” and “probiotic” worldwide from January 2004 through June 2019, and the time series decomposition analysis by Loess. The *x*-axes of all of the mini-figures are the same periods. The *y*-axes are expressed in RSV. The mini-figure “observed” presents the course over time of the RSV of the topic. The mini-figure “trend” presents the trend over the analyzed period. The mini-figure “seasonal” presents the seasonal component of the time series “observed”. The mini-figure “irregular” presents the variability, independent from the main trend and seasonal variation.

**Figure 2 antibiotics-08-00147-f002:**
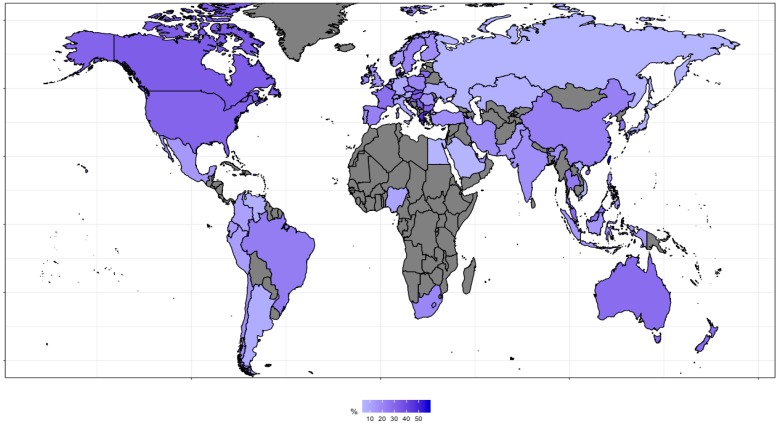
The relative interest of the topic “probiotics” worldwide. Gray color represents countries with low search volume, which were not included in the analysis. The list of the countries and values are available in Dataset in file “Figure 2 & Appendix A”.

**Table 1 antibiotics-08-00147-t001:** Time series analysis of the topics. Relative search volume (RSV) presented as mean ± standard deviation. TBATS—exponential smoothing state space model with Box-Cox transformation, autoregressive-moving average errors, and trend and seasonal components. * *p* < 0.05, ** *p* < 0.01, *** *p* < 0.01.

Topic	Location	RSV	Seasonal Mann–Kendall Test	Slope	TBATS (Seasonality Present, Period [month])	Month with the Lowest Seasonal Component	Month with the Highest Seasonal Component	Seasonal Component Amplitude
Antibiotics	Worldwide	57.5 ± 17.9	tau = 0.88 ***	3.7 RSV/year (6.5%/year)	YES, 12	July (−4.8 RSV)	February (5.0 RSV)	9.8 RSV (17.0%)
Probiotic	Worldwide	14.1 ± 7.9	tau = 0.98 ***	1.7 RSV/year (12.1%/year)	YES, 12	December (−1.6 RSV)	February (1.1 RSV)	2.7 RSV (19.1%)
Antibiotics	Argentina	45.9 ± 20.2	tau = 0.77 ***	3.9 RSV/year (8.5%/year)	YES, 12	February (−5.4 RSV)	June (5.7 RSV)	11.1 RSV (24.2%)
Probiotic	Argentina	4.8 ± 2.8	tau = 0.32 ***	0.2 RSV/year (4.2%/year)	YES, 12	January (−1.4 RSV)	June (1.0 RSV)	2.4 RSV (50.0%)
Antibiotics	Australia	54.5 ± 18.9	tau = 0.86 ***	3.8 RSV/year (7.0%/year)	YES, 12	January (−9.3 RSV)	August (7.2 RSV)	16.5 RSV (30.3%)
Probiotic	Australia	18.3 ± 14.2	tau = 0.88 ***	2.9 RSV/year (15.8%)	YES, 12	December (−3.9 RSV)	September (2.1 RSV)	6.0 RSV (32.8%)
Antibiotics	South Africa	53.7 ± 17.1	tau = 0.65 ***	2.6 RSV/year (4.9%)	YES, 12	December (−7.6 RSV)	August (6.5 RSV)	14.1 RSV (26.3%)
Probiotic	South Africa	14.5 ± 6.7	tau = 0.38 ***	0.5 RSV/year (3.4%)	NO, −	-	-	-

**Table 2 antibiotics-08-00147-t002:** R Spearman rank-correlation results.

	RSV Antibiotics, 2015	RSV Probiotic, 2015	Antibiotic Consumption, 2015	Drug Resistance Index, 2015	Health Expenditure Per Capita, 2014
**RSV probiotic, 2015**	Rs = 0.24 *n* = 64				
Antibiotic Consumption, 2015	Rs = 0.14 *n* = 65	**Rs = 0.35 ***n* = 67**			
Drug Resistance Index, 2015	Rs = −0.13 *n* = 40	Rs = −0.21 *n* = 40	Rs = 0.20 *n* = 40		
Inflation-Adjusted Health Expenditure per Capita, 2015	Rs = 0.09 *n* = 72	**Rs = 0.41 *** *n* = 61**	**Rs = 0.27 * *n* = 63**	**Rs = −0.84 *** *n* = 39**	
Human Development Index, 2015	Rs = 0.09 *n* = 73	**Rs = 0.44 *** *n* = 62**	**Rs = 0.28 * *n* = 63**	**Rs = −0.84 *** *n* = 39**	**Rs = 0.95 *** *n* = 72**

* *p* < 0.05, ** *p* < 0.01, *** *p* < 0.001.

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
