# Peer review of "Global Internet Data on the Interest in Antibiotics and Probiotics Generated by Google Trends"

_antibiotics, 2019, doi:10.3390/antibiotics8030147_

Round 1
Reviewer 1 Report
Manuscript covered a well-time issue and analyses yearly trends and seasonal variation in Google searches associated with the ‘antibiotics’ and ‘probiotic’ from 14 January 2004 to June 2019. This study is novel in the sense that the present study reported the interest of global internet users in antibiotics, probiotics prebiotics, synbiotics, and antimicrobial resistance. It has been written in a clear and comprehensive way and reflects the purpose of the study. However, some minor edits may improve the quality of the article.
The authors should consider adding the name of the country in Figure 2 for reader convenience. line # 140 Author should consider mentioning the associations in 2-3 line instead of writing …. found several significant associations ….. It will be good to define Drug Resistance Index, Inflation-adjusted Health Expenditure per capita, Human Development Index in the method section before showing association in Table 2. Line # 80-82 The sentence “Because most Google users live in the northern hemisphere, we chose three countries 80 (Argentina, Australia, and South Africa) from the southern hemisphere to perform time series decomposition analysis on separately, to reveal possible different RSV seasonal patterns.” is not clear please rewrite the sentence Line # 40, 64, 109, 158, 175, 190 and 206: There is a spacing problem between the sentence. Table 2 Expedinture should be corrected as Expenditure
Author Response
Response to reviewers’ comments
We thank the reviewer for the very careful evaluation of our manuscript and we are grateful
for valuable comments. We have uploaded the most recent version of the manuscript.
All changes are highlighted. We were encouraged to address the following points:
We have been advised to check the manuscript by a native. We previously sent the manuscript for professional language editing. Nevertheless we have analyzed the text again and found several errors picked up by Reviewer. All these errors have been amended.
Reviewer #1
The authors should consider adding the name of the country in Figure 2 for reader convenience.
The name of the countries in Figure 2 are presented in Dataset in table "figure 2 & figure s1.xls".
Therefore we propose adding to figure caption the following sentence:
„The list of the countries and values are available in Dataset in file „figure 2 & figure s1.xls”.”
Alternatively, we may add Supplementary Table contaning the list of countries and following description:
„The list of the countries and values are available in Table SX”
line # 140 Author should consider mentioning the associations in 2-3 line instead of writing …. found several significant associations …..
We edited the sentences as follows:
We found significant associations between the RSVs of ‘probiotic’ and antibiotic consumption, inflation-adjusted health expenditure per capita and HDI (Table 2). There were no significant associations for ‘antibiotics’.
It will be good to define Drug Resistance Index, Inflation-adjusted Health Expenditure per capita, Human Development Index in the method section before showing association in Table 2.
We edited the method section.
DRI represents combined ability of antibiotics to treat infections with the extent of their use in clinical circumstances for a specific region [19]. Inflation-adjusted health expenditure per capita is a universal measure of sum of private and public health funding. The outcome is adjusted to 2011 dollar. Human Development Index is combined index of life expectancy, education, and per capita income that allow comparing the development of the countries.
Line # 80-82 The sentence “Because most Google users live in the northern hemisphere, we chose three countries 80 (Argentina, Australia, and South Africa) from the southern hemisphere to perform time series decomposition analysis on separately, to reveal possible different RSV seasonal patterns.” is not clear please rewrite the sentence
We propose the following sentence.
Most Google users live in the northern hemisphere thus most of them experience cold season from December to February. We chose three countries (Argentina, Australia, and South Africa) from the southern hemisphere to perform additional time series decomposition, to reveal possible inverse RSV seasonal patterns related to difference in months of cold season.
Line # 40, 64, 109, 158, 175, 190 and 206: There is a spacing problem between the sentence.
We corrected the errors.
Table 2 Expedinture should be corrected as Expenditure
Corrected.
Reviewer 2 Report
This research deals with Google Trends data regarding antibiotics and probiotics. Although the results showed seasonal and regional differences in antibiotic- and probiotic-related information, the tendency can be expected and the significantly higher interest on RSV search of antibiotics topics did not show the correlations with antibiotic consumption, Drug Resistance Index, Inflation-adjusted Health Expedinture per capita, Human Development Index.. etc. The reason why the interest on 'Antibiotics' search should be discussed more specifically in aspects of social and environmental factors.
Author Response
Response to reviewers’ comments
We thank the reviewer for the very careful evaluation of our manuscript and we are grateful
for valuable comments. We have uploaded the most recent version of the manuscript.
All changes are highlighted. We were encouraged to address the following points:
We have been advised to check the manuscript by a native. We previously sent the manuscript for professional language editing. Nevertheless we have analyzed the text again and found several errors picked up by Reviewer. All these errors have been amended.
Reviewer #2
This research deals with Google Trends data regarding antibiotics and probiotics. Although the results showed seasonal and regional differences in antibiotic- and probiotic-related information, the tendency can be expected and the significantly higher interest on RSV search of antibiotics topics did not show the correlations with antibiotic consumption, Drug Resistance Index, Inflation-adjusted Health Expedinture per capita, Human Development Index.. etc. The reason why the interest on 'Antibiotics' search should be discussed more specifically in aspects of social and environmental factors.
We edited the discussion as follows:
Global antibiotic consumption increased by 39% between 2000 and 2015, and it is estimated that it will be up to 200% higher in 2030 than in 2015 [25]. The probiotics market is growing and is speculated to grow 37% globally from 2016 to 2020 [26]. The RSV of both ‘antibiotic’ and ‘probiotic’ increases over time. This might be associated with the growing consumption of both types of products. Additionally, the growing number of studies and the regulation of antibiotics and probiotics may generate public discourse. However, since 1990 no new class of antibiotics were discovered. We found that the RSV of the topic ‘probiotics’ increases relatively twice as fast as ‘antibiotics’. This may be related to the wide range of new beneficial effects of probiotics in the prevention of surgery-related complications [27], mental health [28], cardiovascular system [29,30], insulin resistance [31], atopy [32], and many others. Furthermore, due to regulations probiotics can be publicly advertized, while antibiotics cannot. It is also suggested that community is aware on antibiotics but have a little knowledge on properties, risk and antimicrobial resistance [33]. (...)
Interestingly, the RSV of the topic ‘antibiotics’ was not significantly associated with antibiotic consumption. The interest of ‘antibiotics’ may not mirror the antibiotic consumption due to health policy limiting access to this type of drugs. Therefore, the interest and will of a Google user cannot be easily converted into the purchase. Moreover, antibiotic consumption is highly dependent on in-hospital use of antibiotics [36,37], we hypothesize that hospitalized patients may not generate Internet traffic related to the topic ‘antibiotics’. Antibiotics represent the old type of drugs and are used by clinicians all over the world. Therefore, the interest in the topic ‘antibiotics’ may not correspond with health expenditure per capita, and HDI. DRI was not significantly associated with interest in ‘antibiotics’. It may be explained by no relation between RSV of the topic and antibiotic consumption which could potentially generate an increase of antibiotic resistance. However, the analysis was performed only for 2015 which limits the conclusions.
Reviewer 3 Report
The authors analyzed world trends in searches for the topics "antibiotics" and "probiotics".
My major comment is that the authors should show data from the USA, Canada and Europe for relative search volume and how it varies across seasons since these countries have a very high relative interest for "probiotics".
Author Response
Response to reviewers’ comments
We thank the reviewer for the very careful evaluation of our manuscript and we are grateful
for valuable comments. We have uploaded the most recent version of the manuscript.
All changes are highlighted. We were encouraged to address the following points:
We have been advised to check the manuscript by a native. We previously sent the manuscript for professional language editing. Nevertheless we have analyzed the text again and found several errors picked up by Reviewer. All these errors have been amended.
Reviewer #3
My major comment is that the authors should show data from the USA, Canada and Europe for relative search volume and how it varies across seasons since these countries have a very high relative interest for "probiotics".
To answer the comment we propose adjusting supplementary material with time series decomposition from several countries with the highest relative interest for „probiotics”:
in Methods:
Additionally, we analyzed time series decomposition of RSV of several countries with the highest relative interest for „probiotics”.
In Results:
We analyzed the time series decomposition of RSV of Taiwan, Greece, Canada and the United States (Table S2, Figure S4A-D).
In table/figures captions:
Table S2
Additional the time series analysis of the topics. Relative Search Volume presented as mean ± standard deviation.
* p < 0.05, ** p < 0.01, *** p < 0.001
NS – Non significant; RSV – Relative Search Volume, TBATS - exponential smoothing state space model with Box-Cox transformation, autoregressive-moving average errors, and trend and seasonal components
Figure S4A
Relative search volume (RSV) of topics „antibiotic” and „probiotic” in Canada from January 2004 through June 2019 and time series decomposition analysis by Loess.
X-axes of all mini-figures are the same periods. Y-axes are expressed in RSV.The mini-figure „observed” presents course over time of RSV of the topic. The mini-figure „trend” presents the trend over the analyzed period.The mini-figure „seasonal” presents the seasonal component of the time series „observed”.The mini-figure „irregular” presents variability independent from the main trend and seasonal variation.
Figure S4B
Relative search volume (RSV) of topics „antibiotic” and „probiotic” in Greece from January 2004 through June 2019 and time series decomposition analysis by Loess.
X-axes of all mini-figures are the same periods. Y-axes are expressed in RSV.The mini-figure „observed” presents course over time of RSV of the topic. The mini-figure „trend” presents the trend over the analyzed period.The mini-figure „seasonal” presents the seasonal component of the time series „observed”.The mini-figure „irregular” presents variability independent from the main trend and seasonal variation.
Figure S4C
Relative search volume (RSV) of topics „antibiotic” and „probiotic” in Taiwan from January 2004 through June 2019 and time series decomposition analysis by Loess.
X-axes of all mini-figures are the same periods. Y-axes are expressed in RSV.The mini-figure „observed” presents course over time of RSV of the topic. The mini-figure „trend” presents the trend over the analyzed period.The mini-figure „seasonal” presents the seasonal component of the time series „observed”.The mini-figure „irregular” presents variability independent from the main trend and seasonal variation.
Figure S4D
Relative search volume (RSV) of topics „antibiotic” and „probiotic” in The United States from January 2004 through June 2019 and time series decomposition analysis by Loess.
X-axes of all mini-figures are the same periods. Y-axes are expressed in RSV.The mini-figure „observed” presents course over time of RSV of the topic. The mini-figure „trend” presents the trend over the analyzed period.The mini-figure „seasonal” presents the seasonal component of the time series „observed”.The mini-figure „irregular” presents variability independent from the main trend and seasonal variation.
Furthermore, we expanded the attached Dataset.
Round 2
Reviewer 3 Report
The major concern raised has been addressed by the authors.
However, the supplementary figures that have been included (labeled as Supplementary Figure 3 A-D) have been wrongly referred to as Supplementary Fig 4A-D in the author's reply. Please ensure that the figure legends have been correctly labeled in the main text.